# Association of viral kinetics, infection history, NS1 protein with plasma leakage among Indonesian dengue infected patients

**Leonard Nainggolan**[1]*, **Beti Ernawati Dewi**[2], **Arif Hakiki**[1], **Antony Joseph Pranata**[2], **Tjahjani Mirawati Sudiro**[2], **Byron Martina**[3], **Eric van Gorp**[3]

**1** Division of Tropical and Infectious Disease, Department of Internal Medicine, Cipto Mangunkusumo Hospital, Faculty of Medicine Universitas Indonesia, Jakarta, Indonesia, **2** Department of Microbiology, Faculty of Medicine Universitas Indonesia, Jakarta, Indonesia, **3** Department of Viroscience, Erasmus Medical Center, Rotterdam, The Netherlands

* leonard.nainggolan@ui.ac.id, leonard.golan@yahoo.com

**Data Availability Statement:** All relevant data are within the paper and its Supporting Information files.

## Abstract

### Objectives

Plasma leakage, a hallmark of disease in Dengue virus (DENV) infection, is an important clinical manifestation and is often associated with numerous factors such as viral factors. The aim of this study is to investigate the association of virus serotype, viral load kinetics, history of infection, and NS1 protein with plasma leakage.

### Methods

Subjects with fever $\leq$ 48 hours and positive DENV infection were included. Serial laboratory tests, viral load measurements, and ultrasonography examination to assess plasma leakage were performed.

### Results

DENV-3 was the most common serotype found in the plasma leakage group (35%). Patients with plasma leakage demonstrated a trend of higher viral load and a longer duration of viremia compared to those without. This was significantly observed on the fourth day of fever (p = 0.037). We found higher viral loads on specific days in patients with plasma leakage in both primary and secondary infections compared to those without. In addition, we also observed more rapid viral clearance in patients with secondary infection. NS1 protein, especially after 4 days of fever, was associated with higher peak viral load level, even though it was not statistically significant (p = 0.470). However, pairwise comparison demonstrated that peak viral load level in the group of patients with circulating NS1 detected for 7 days was significantly higher than the 5-day group (p = 0.037).

### Conclusion

DENV-3 was the most common serotype to cause plasma leakage. Patients with plasma leakage showed a trend of higher viral load and a longer duration of viremia. Higher level of viral load was observed significantly on day 5 in patients with primary infection and more

**Funding:** The study was financially supported by the Indonesian Science Fund grant (08/DIPI/2017-MR/PO17509/1), Indonesian Ministry of Education, Culture, Research, and Technology grant (PUTI Q2 582/UN2.RST/HKP.05.00/2022) and internal grants from the Department of Internal Medicine and the Department of Microbiology, Faculty of Medicine Universitas Indonesia. The funders had no role in study design, data collection, data analyses, decision to publish or preparation of the manuscript."

**Competing interests:** The authors affirm no conflict of interest in this study.

**Abbreviations:** DENV, : dengue virus; DF, dengue fever; DHF, dengue hemorrhagic fever; qRT-PCR, quantitative reverse transcriptase real time polymerase chain reaction; NS1, nonstructural 1; WHO, World Health Organization.

rapid viral clearance was observed in patients with secondary infection. Longer duration of circulating NS1 protein was also seen to be positively correlated with higher peak viral load level although not statistically significant.

## Introduction

Dengue infection is a major health problem in Indonesia and other tropical and subtropical countries. Dengue virus (DENV) infection consisted of four serotypes, namely DENV-1, DENV-2, DENV-3, and DENV-4. Since the first case report in 1969, all four DENV serotypes have been circulating in Indonesia and undergone shifts in dominance [1].

DENV infection is typically a self-limiting disease with a mortality rate of less than 1% if treated properly. DENV causes a wide spectrum of disease ranging from asymptomatic infection, dengue fever (DF), dengue hemorrhagic fever (DHF), and dengue shock syndrome (DSS) which is potentially life-threatening form of the disease. Plasma leakage is one of the manifestations of severe DENV infection and has been widely used as the criteria to differentiate DF from DHF. Furthermore, extensive plasma leakage can lead to hypoperfusion, shock, and eventually death [2, 3].

The pathogenesis of DENV infection is still not fully understood due to the lack of suitable animal model that emulates dengue disease, especially the severe forms of the disease (DHF/DSS). This has hindered progress in many areas of DENV research [4, 5]. Viral load has been hypothesized to play an important role in the disease pathogenesis. Several studies observed a positive correlation between viral load and disease severity [6–9]. It has also been demonstrated that infection with a certain serotype can lead to a more severe disease manifestations [9–12].

To date, the association between viral factors and plasma leakage is still not fully elucidated and there is a limited human study that provides viral load kinetics information from the early days of infection. The aims of this study were to identify the virus serotype, viral load kinetics, infection history, NS1 protein and their association with plasma leakage in the early days of disease of a cohort of dengue patients in Indonesia.

## Materials and methods

### Study design and population

This prospective observational study was conducted in Jakarta, Indonesia from February 2010 to January 2011. We carried out active case finding by three means; through recruitment at Community Health Centers ("Puskesmas"), examination by general practitioners in several primary health care facilities, and a hotline number which can be contacted by subjects meeting the inclusion criteria. The inclusion criteria were patients aged >14 years with fever for ≤48 hours, having typical clinical manifestations of dengue such as headache, retro-orbital pain, myalgia, arthralgia, rash, hemorrhagic manifestation, and willing to participate in this study. Expectant mother, patients suffering from immunological disease, heart failure, chronic kidney disease, or liver cirrhosis were excluded. Subjects were recruited consecutively.

We subsequently performed DENV nonstructural protein 1 (NS1) antigen rapid test (SD Dengue Duo, Standard Diagnostics, Republic of Korea) on patients that fulfilled the inclusion criteria. Patients with positive result of DENV NS1 antigen test were then hospitalized at Cipto Mangunkusumo Hospital to undergo further treatment and daily observation until they recovered. Reverse-transcriptase polymerase chain reaction (RT-PCR) were performed to confirm

**Table 1. Subjects' characteristics.**

| Variable | No plasma leakage | Plasma leakage | P-value |
|---|---|---|---|
| All subjects (%) | 28 (58.3) | 20 (41.7) | |
| Sex | | | |
| Men (%) | 14 (29.2) | 10 (20.8) | |
| Women (%) | 14 (29.2) | 10 (20.8) | |
| Dengue classification | | | |
| DF | 28 | 0 | |
| DHF | 0 | 20 | |
| DSS | 0 | 0 | |
| Dengue infection history | | | |
| Primary (%) | 11 (22.9) | 4 (8.3) | |
| Secondary (%) | 17 (35.4) | 16 (33.3) | |
| Complete blood count | | | |
| Hemoglobin (g/dl), mean (±SD) | 13.9 (1.34) | 13.8 (1.23) | P = 0.877 |
| Hematocrit (%), mean (±SD) | 43.8 (3.31) | 41.9 (3.33) | P = 0.937 |
| Leukocyte (μl), median (min-max) | 3948 (2870–4320) | 3715 (2270–8930) | P = 0.189[#] |
| Platelets (cells/mm$^3$), mean (±SD) | 139,150 (32,149) | 129,962 (49,822) | P = 0.091 |
| AST levels (u/l), median (min-max) | 33.0 (17–173) | 31.5 (19–108) | P = 0.841[#] |
| ALT levels (u/l), median (min-max) | 22.0 (7–106) | 20.5 (7–141) | P = 0.777[#] |
| Albumin (mg/dl), median (min-max) | 4.3 (3.8–4.7) | 4.1 (3.4–4.5) | P = 0.481[#] |

Complete blood count results were from day 2 of fever

*Between plasma leakage+ and no plasma leakage groups. Unpaired t test was used unless otherwise stated; #Mann Whitney U test.

the diagnosis while LightMix Dengue real-time PCR assays were performed to assess viral load levels. Disease severity was classified using the 1997 WHO classification system (**Table 1**).

## Clinical and laboratory parameters

The subjects' characteristics and baseline information were reported on a standardized case report form. Subjects were divided into two groups based on the presence of plasma leakage. Patients who showed an increase of haematocrit values ≥ 20% from baseline or decrease in convalescence and/or pleural effusion and/or ascites and/or hypoalbuminaemia were categorized into plasma leakage patient [3]. Complete blood count was performed every 12 hours and quantitative real-time polymerase chain reaction (qRT-PCR) examination was performed every 24 hours until the sixth day of the fever. Furthermore, ultrasonography examination was also performed on day 3, 5, and 7 of fever to detect ascites and/or pleural effusion as indicators for plasma leakage. IgM and IgG dengue antibody tests were performed from the first day until the seventh day of the fever to determine whether the patient had primary or secondary infection. A positive IgM dengue antibody test with negative IgG was considered as primary infection, while positive IgM and IgG dengue antibody test or IgG alone was considered as secondary dengue infection. Hemagglutination inhibition test was also performed to confirm the history of infection as mentioned in the study by Dewi et al. [12].

## Viral load and serotype determinations

RNA was extracted from 140 μl of plasma using QiAmp viral RNA kit (Qiagen, USA). Dengue virus serotype was determined with Lanciotti's two-step reverse transcriptase PCR with slight modification [12, 13]. The first step PCR mixture contained 4μL of 10x PCR buffer with

1.5mM of MgCl2, 0.8μL of 10 μM Land D1 and Land D2 primers, 0.8 μL of each 10 mM dNTP, 0.4 μL of SuperScript II RTase (Invitrogen, Carlsbad, CA, USA)., 0.15 μL of 5 U/μL Platinum *Taq* DNA polymerase (Invitrogen, Carlsbad, CA, USA)., and 8 μL of RNA. Thermocycling was conducted with a sequence of 53˚C for 30 min, 95˚C for 5 min; 30 cycles at 95˚C for 45s, 60˚C for 30s, and 72˚C for 90 s; and a final heating at 72˚C for 7 min. The second PCR mixture contained 2.5 μL of 10 PR buffer with 1.5 mM MgCl2, 0.5 μL of 10 mM each dNTP, 0.15 μL of 5 U/μL of Platinum *Taq* DNA polymerase (Invitrogen, Carlsbad, CA, USA), 1 μL of 10 μM D1, TS1, TS2, TS3 and TS4 primers, and 2 μL of the first PCR product. Thermocycling was conducted with a sequence of 95˚C for 5 min; 35 cycles at 95˚C for 45 s, 60˚C for 30 s, and 72˚C for 60 s; and a final step at 72˚C for 7 min.

Serotype specific PCR was performed on the initial blood samples of each patient. Complementary DNA (cDNA) was prepared using random hexamer primer in Transcriptor 1[st] strand cDNA Synthesis Kit (Roche Diagnostic, Switzerland). Subsequently, viral RNA quantification was determined by qRT-PCR using LightCycler FastStart DNA Master Hybridization Probes and LightMix Kit dengue virus (Roche Diagnostic, Switzerland). The experiments and quantification of virus were performed according to the manufacturer's protocol. The viral load level was presented in copies/ml.

### Kinetic of NS-1 protein

To analysis correlation between NS1 protein and viral load, NS1 antigen was detected by NS1 rapid test (Standard Diagnostic, Kyonggi, Korea) from the first day to the seventh day of illness.

### Statistical analysis

The data were analyzed using SPSS version 20.0 and GraphPad Prism version 9.0 for Windows. Categorical variables such as sex, age, day of fever, type of infection, the presence of NS-1 antigen and the dengue serotype were descriptively presented in numbers and percentages. Numerical variables, i.e. serial dengue viral load were presented in mean with standard deviation and 95% confidence interval or in median with interquartile range. A bivariate analysis with either Student's t-test or Mann-Whitney U test was performed to determine the association between dengue viral load and the occurrence of plasma leakage. Kruskal-Wallis' test was performed to analyze the association between NS1 positivity and viral load levels.

### Ethics

Ethical clearance was obtained from the Health Research Ethics Committee, Faculty of Medicine Universitas Indonesia No. 71/PT02.FK/ETIK/2009 and written informed consent was obtained from all study participants. Written informed consent was also obtained from parents or legal guardians for participants aged < 18 years. Data collection and tests were carried out in accordance with the relevant guidelines and regulation in respect of human rights and the privacy of each participant.

## Results

### Subject characteristics

Our study enrolled 53 patients who met the inclusion criteria. Forty-eight patients who had positive RT-PCR results were included for further analyses. Male and female patients were proportional in numbers; with mean age 24.3 years. Seventeen patients included in this study were admitted on their first day of fever while the rest of the patients were admitted on the

second day of fever. The duration of fever ranged from 2 to 7 days, with the mean of 4.1 days. Twenty patients (41.7%) had plasma leakage. Those subjects with plasma leakage were categorized as DHF, in which 9 patients were categorized as DHF grade 1 and the other 11 patients were classified as DHF grade 2 based on the 1997 WHO classification. Thirty-three subjects (68.75%) were categorized as having a secondary infection. Primary dengue infected patients had fever for an average duration of 5 days while secondary dengue patients had an average duration of 4 days. All patients recovered from the infection and discharged from the hospital by the end of the treatment. **Table 1** shows the characteristics of the patients.

## Dengue virus serotypes and its association with plasma leakage

Dengue virus 2 was the predominant serotype found in this study, which was found in 19 subjects (39.6%), followed by DENV-1 (14 subjects; 29.2%). DENV-2 was the most frequent serotype in both primary and secondary infection. In DENV-1 and DENV-2 infected patients, 64.29% and 68.42% of patients had mild diseases without plasma leakage, respectively. Similar result was found in DENV-4 infected patients (**Table 2**). In contrast, 58.33% of DENV-3 infected patients showed more severe disease (DHF-I and DHF-II). Even though plasma leakage proportion is more common in DENV-3 infected patients, there was no statistical difference observed between serotypes. (**Table 2**).

## Dengue viral load during treatment and its association with plasma leakage

Based on the qRT-PCR examination, the viral load levels in the group with plasma leakage were higher than the group without plasma leakage throughout the study period. The difference of dengue viral load between both groups was statistically significant on the fourth day of the fever (p = 0.024) (**Table 3**).

## Dengue viral load based on history of infection

In patients with secondary infection viral load level were the highest on the first day of the fever and reached its lowest level on the fifth day of the fever (**Fig 1**). The highest level of viral load in primary infection was detected on the third day of illness and then decreased slower compared to that in the secondary infection and reached lowest level on the sixth day of illness. We further compared the viral load between patients with and without plasma leakage based

**Table 2. DENV serotypes and its association with plasma leakage.**

| Variable | Without plasma leakage (%) | With plasma leakage (%) | Total (%) | P values |
|---|---|---|---|---|
| **Dengue Serotype** | | | | |
| DENV-1 | 9 (18.7) | 5 (10.4) | 14 (29.2) | 0.06[a]/0.25[b]/0.57[c] |
| DENV-2 | 13 (27.1) | 6 (12.5) | 19 (39.6) | 0.14[d]/0.53[e] |
| DENV-3 | 5 (10.4) | 7 (14.6) | 12 (25.0) | 1[f] |
| DENV-4 | 1 (2.1) | 2 (4.2) | 3 (6.2) | - |

Note

[a]DENV-1 vs DENV-2

[b]DENV-1 vs DENV-3

[c]DENV-1 vs DENV-4

[d]DENV-2 vs DENV-3

[e]DENV-2 vs DENV-4

[f]DENV-3 vs DENV-4. Two-step qRT-PCR was conducted to determine the virus serotypes. Chi-square or Fischer exact test was used to determine the association between DENV serotype and plasma leakage.

**Table 3. Dengue viral load median score difference between subjects with and without plasma leakage based on the day of fever.**

| Day of fever | Median of Viral Load | | Mann-Whitney test (p value) |
|---|---|---|---|
| | Without plasma leakage | With plasma leakage | |
| Day 1 | $5 \times 10^5$ (23,5–188,5$\times 10^5$) n = 5 | $6.7 \times 10^5$ (35.25–182.5$\times 10^5$) n = 5 | 0.548 |
| Day 2 | $2.8 \times 10^5$(0–$5.6 \times 10^7$) n = 28 | $14.4 \times 10^5$ (0–$1.34 \times 10^8$) n = 20 | 0.146 |
| Day 3 | $1.28 \times 10^5$(0–$46.25 \times 10^5$) n = 28 | $18.85 \times 10^5$ (17–$1.38 \times 10^8$) n = 20 | 0.066 |
| Day 4 | $3.3 \times 10^3$ (0–$14.2 \times 10^6$) n = 28 | $24.89 \times 10^5$ (0–$2.33 \times 10^8$) n = 20 | **0.024**\* |
| Day 5 | 0 (0–$3.35 \times 10^5$) n = 28 | 210 (0–$1.39 \times 10^8$) n = 20 | 0.176 |
| Day 6 | 0 (0–$5.25 \times 10^5$) n = 23 | 0 (0–$4.6 \times 10^3$) n = 16 | 0.207 |

The unit of median scores of viral load was copies/ml. Data are presented as median with interquartile range
\*: statistically significant.

on the infection history (**Fig 2**). The viral load levels of the patients with plasma leakage were higher than the group without plasma leakage throughout the study duration in patients with primary infection, except on the first day of fever (**Fig 2A**). The viral load was significantly higher on day 4 (P = 0.010) of the fever in the leakage group when we compared patients with and without plasma leakage in the primary infection group. In patients with secondary infection, the viral load levels of plasma leakage patients were also higher compared to patients without plasma leakage especially on day 1, 3, and 4, although the differences were not statistically significant (**Fig 2B**).

## NS1 positivity during course of illness

The number of patients with positive NS1 was the highest in the first 24 hour of illness and decreasing throughout the course of illness. The NS1 positivity was significantly reduced between the day 3 and day 4 interval and between the day 4 and day 5 interval (**Fig 3A**). Moreover, the proportion of patients with the highest NS1 positivity was shown during the early days of illness and the proportion of plasma leakage group was seen to be markedly reduced compared to the group without plasma leakage during the later phase of illness (**Fig 3B**).

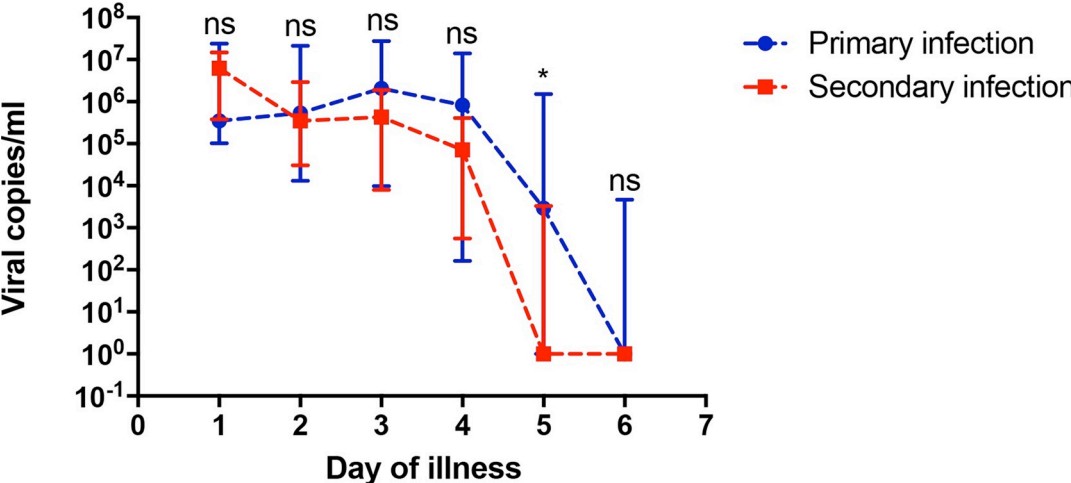

**Fig 1. Plasma dengue viral load in patients based on the infection history.** Significant difference between patients with primary infection and secondary infection was observed on day 5 (P = 0.033). Data are presented as median with interquartile range (IQR). Abbreviations: ns, not significant.

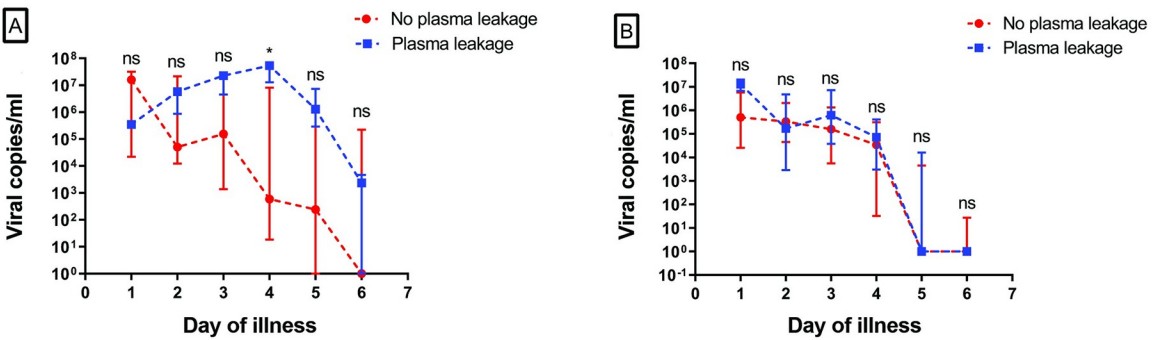

**Fig 2. Plasma dengue viral load in patients with and without plasma leakage based on the infection history.** Plasma viral load was compared between patients with and without plasma leakage in primary (A) and secondary (B) infections. (A) Viral load levels are significantly increased in plasma leakage patients compared to no plasma leakage group on day 4 (P = 0.010). (B) Viral load level is comparatively higher in day 1, 3 and 4 in plasma leakage patients, albeit not significant. Data are presented as median with interquartile range (IQR). *Represents a p value < 0.05; **p value < 0.01; ***p value < 0.001. Abbreviations: ns, not significant.

### Peak viral load levels based on circulating NS-1 duration

In this analysis, viral load levels were taken from the peak viral load observed throughout the study. Viral load levels were then compared based on the duration of NS1 positivity. Kruskal-Wallis' test showed no significant differences between peak viral load levels throughout the study period (p = 0.470).

Despite NS1 was normatively tested during day 1–4 of the fever in this study, we analyze the presence of circulating NS1 after day 4 of illness. Patients whose NS1 was still detected after 4 days were grouped and the peak viral load levels were analyzed. Kruskal-Wallis' test revealed that the longer NS1 were detected, the higher viral load levels were seen, although the differences were not statistically significant (Fig 4). However, further unpaired T-test comparison revealed that the peak levels between the 5-day group and 7-day group differed significantly (p = 0.037).

Although the peak viral load levels were positively associated with the longer duration of circulating NS1, comparison of viral load levels and NS1 differed especially in plasma leakage group, showing faster viral load clearance in the plasma leakage group compared to NS1 (Fig 5A and 5B).

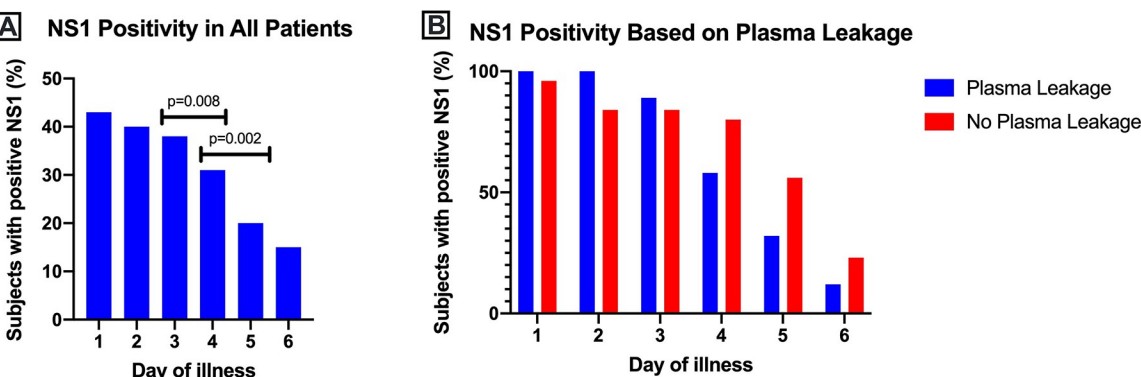

**Fig 3. NS1 positivity during course of illness.** NS1 positivity was analyzed (A) in all patients and (B) based on the occurrence of plasma leakage. Data are presented as proportion of subjects.

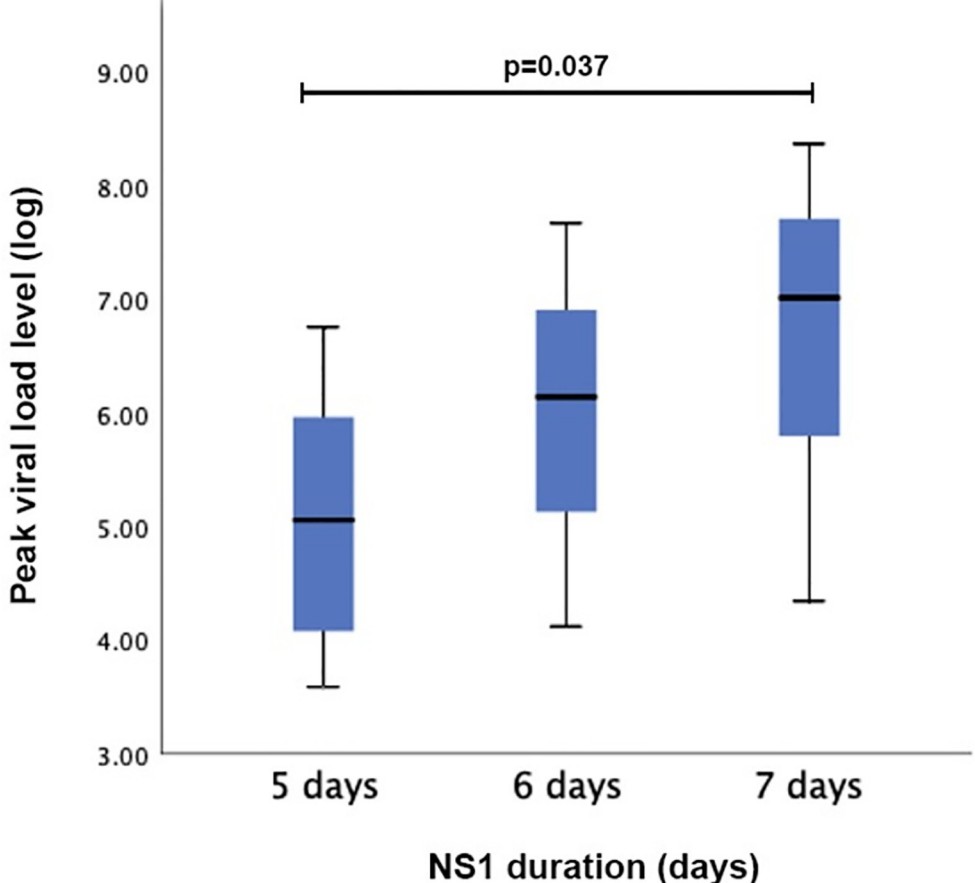

**Fig 4. Peak viral load based on the duration of circulating NS1 after 4 days.** Kruskall-Wallis' test showed no significant difference between peak viral load levels in different duration of circulating NS1, however after 4 days, higher peak of viral load seem to be correlated with the longer duration of NS1. The peak levels between the 5-day group and 7-day group differed significantly (p = 0.037). Viral load levels are presented as log viral load.

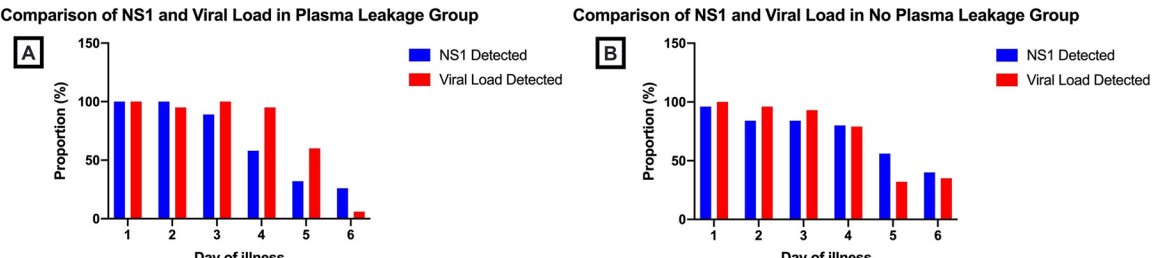

**Fig 5. Comparison of the NS1 positivity and detected viral load.** NS1 positivity and detected viral load was analyzed head-to-head in (A) plasma leakage group and (B) no plasma leakage group. Data are presented as a proportion of subjects.

## Discussion

It has been postulated that different dengue serotypes may have different abilities to cause severe dengue disease [14]. This study revealed that even though DENV-2 was the predominant serotype, DENV-3 was the major serotypes detected in dengue patients with plasma leakage, albeit not significantly correlated with plasma leakage. Severe forms of DENV infection are often considered as a pediatric disease in southern Asia and graded as DHF to DSS [15]. However, we found 41.67% of adult patients in this study with plasma leakage manifestations. This suggest that disease is affecting older populations as other studies indicate in other countries [15].

Previous study in Jakarta also found DENV-2 as the most prevalent serotype, followed by DENV-3 [16]. The study also showed that more DENV-2-infected patients showed more severe disease manifestations although the rate was almost similar with DENV-3. A serotype switch from DENV-2 to DENV-3 has been postulated to play a role in the severity outbreak [17]. In Singapore and Malaysia, there was an increased risk of severe disease correlated with DENV-2 and/or DENV-3 infection [18, 19]. The differences in serotypes may occur due to the presence of different genotypes in these regions; and serotype proportion can vary depending on the region, time, and sampling method in which the studies were conducted [20].

In this study, DENV-1/DENV-4 co-infection case was detected. DENV co-infection is commonly found in a hyperendemic region as demonstrated from a previous report that included patients from Indonesia, Mexico, and Puerto Rico. The study showed a total of 16 out of 292 samples had dengue co-infections, 11 patients of which were from Indonesia [21]. In Malaysia, 15.3% of dengue positive cases had concurrent infections by different DENV serotypes. This study [22] also showed that patients infected with more than one serotype had a more severe clinical manifestations. However the patient in our study neither developed severe clinical manifestations nor complications. Nevertheless, we cannot further conclude this finding as we only found one case in our study.

Our study also found that plasma leakage group had higher viral load levels compared to nonplasma leakage group and the difference was significantly observed on the fourth day of fever. This is in line with other studies showing that DHF or plasma leakage group had higher viral load and longer duration of viremia compared to DF or group without plasma leakage [9, 23, 24]. Active viral replication and delay in the clearance of viremia were hypothesized to be the factors that contribute to higher viremia in plasma leakage group [8].

Secondary dengue infection remained as a risk factor of higher severity due to the presence of pre-existing cross-reactive and sub-neutralizing antibodies, known as antibody-dependent enhancement (ADE). Opsonization in ADE promotes the entry of virus particles into cells, increasing the number of virus replication and the enhancement of immunopathology [20, 25–30]. Interestingly, this study found a significant difference in viral load levels between primary and secondary infection on the fifth day, with the primary infection group having higher viral load levels. Specific DENV serotypes have been hypothesized to play a role in this finding [30, 31]. Subsequent infections caused by DENV-1 or DENV-2 have no correlation with clinical severity or viral burden [30, 31]. Moreover, this study showed faster viral clearance in patients with secondary infection compared to primary infection as consistently recorded previously [32, 33]. The prompt reaction of the adaptive immune response during secondary dengue infection is dominated by memory B and T cells and some components of this immune arm mediate a robust anti-viral response that may explain rapid viral clearance in secondary infection [2]. NS1, another important viral marker, was also found to be circulating longer in primary infection in comparison to secondary infection [34]. While this finding may imply more rapid viral clearance in secondary infection, further investigation is still further needed. Patients with primary infection also had longer duration of fever that lasted for 5 days,

compared with secondary infection which lasted for 4 days. It is possible that duration of fever was influenced by the duration of viremia since DENV infection leads to production of pyrogenic mediators [2].

The comparison of viral load kinetics between the plasma leakage and non-plasma leakage group in the perspective of primary and secondary infection is also taken into consideration in this study. Three points may be noted from our study. Firstly, viral load levels were higher in plasma leakage group in both primary and secondary infection. Higher viral loads in DHF compared to DF patients both in primary and secondary infection has been reported previously [33]. This finding consistently suggests that viral load levels may contribute to plasma leakage regardless of the infection history. Secondly, in primary infection (Fig 2A), viral load levels between non-plasma leakage group and plasma leakage group differed significantly on the fourth day of fever with the latter having longer duration of viremia. Previous studies supported this result, with higher level of viral loads near defervescence day in dengue with higher severity [6, 8, 23]. Longer duration of viremia in the plasma leakage group suggests an impaired T-cell response in acute dengue infection contributing to the disease severity [26, 35]. Thirdly, rapid viral load clearance is clearly observed in secondary infection regardless of the occurrence of plasma leakage.

In this study, circulating NS1 protein was also associated to the occurrence of plasma leakage. NS1 protein was still detected even after fourth day of illness in both no plasma leakage and plasma leakage groups. With the latter being seen to have NS1 reduced at a faster rate. Patients with DHF have been shown to have faster NS1 clearance [33], implying the role of secondary infection in plasma leakage, viral, and antigenemia elimination [33, 36]. Further observation in our study showed that after fourth day of illness, NS1 which were still circulating, was positively associated with higher peak of viral load level, albeit the result was not statistically significant. However, comparison between 5-day group and 7-day group of the circulating NS1 duration showed that the peak viral load level significantly differed (p = 0.037). This finding was supported with the fact that higher viremia is positively correlated with NS1 positivity [8, 37], especially in the advanced stage of illness [38]. However comparison between the detection of NS1 and viral load revealed that viral load was seen to be reducing faster than NS1. This finding suggests that while longer duration of circulating NS1 was associated with higher peak of viral load, different mechanism may be involved in the clearance of viremia and antigenemia [8, 23, 33, 36, 39], denoting that both viral load and NS1 are important and unexchangeable severity markers in dengue infection.

Collectively, this study suggests that there is an association between viral load levels, infection history, NS-1 protein and plasma leakage as a hallmark of disease in dengue infection. Early assessment of viral load, since the early days of infection may improve our knowledge of viral load kinetics in understanding dengue infection.

## Conclusion

DENV-3 was the main serotype detected in plasma leakage patients in our study. Plasma leakage patients had higher viral load and longer duration of viremia compared to non-plasma leakage patients. Higher viral loads were significantly observed on the fourth day of fever. In the perspective of infection history, primary infection significantly exhibited a higher level of viral load on the fifth day of illness. In addition, secondary infection clearly shows more rapid viral clearance in both plasma leakage and non-plasma leakage group. Longer duration of circulating NS1 protein has positive correlation with higher peak of viral load, albeit not statistically significant. This finding may have an implication in the development of prevention strategies, especially in the development of dengue vaccine.

## Supporting information

**S1 Data.**
(XLSX)

## Acknowledgments

We gratefully acknowledge all the subjects who were willing to participate in our study. We thank Fatih Anfasa, MD and Nadira Prajnasari Sanjaya, MD, for their help in writing and editing this paper.

## Author Contributions

**Conceptualization:** Leonard Nainggolan, Beti Ernawati Dewi.

**Data curation:** Leonard Nainggolan, Arif Hakiki, Antony Joseph Pranata.

**Formal analysis:** Arif Hakiki.

**Methodology:** Leonard Nainggolan, Antony Joseph Pranata.

**Supervision:** Leonard Nainggolan, Beti Ernawati Dewi, Tjahjani Mirawati Sudiro.

**Writing – original draft:** Leonard Nainggolan.

**Writing – review & editing:** Tjahjani Mirawati Sudiro, Byron Martina, Eric van Gorp.

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
