## [Decision Letter · Decision Letter 0]

18 May 2022

PONE-D-21-31549Association of Viral Kinetics, Infection History and Plasma Leakage among Indonesian Dengue Infected PatientsPLOS ONE

Dear Dr. Nainggolan,

Thank you for submitting your manuscript to PLOS ONE. After careful consideration, we feel that it has merit but does not fully meet PLOS ONE’s publication criteria as it currently stands. Therefore, we invite you to submit a revised version of the manuscript that addresses the points raised during the review process.

 Your study and findings are very similar to others that have been published previously, and is therefore considered a replication. For this reason, you must provide a strong scientific rationale to justify your study, and discuss the existing literature in detail to clarify this. In addition, please address the comments from both reviewers. 

We look forward to receiving your revised manuscript.

Kind regards,

Ulrike Gertrud Munderloh, Ph.D.

Academic Editor

PLOS ONE

Journal Requirements:

Reviewers' comments:

Reviewer's Responses to Questions

**Comments to the Author**

1. Is the manuscript technically sound, and do the data support the conclusions?

Reviewer #1: Yes

Reviewer #2: No

2. Has the statistical analysis been performed appropriately and rigorously? 

Reviewer #1: Yes

Reviewer #2: No

3. Have the authors made all data underlying the findings in their manuscript fully available?

Reviewer #1: Yes

Reviewer #2: No

4. Is the manuscript presented in an intelligible fashion and written in standard English?

Reviewer #1: No

Reviewer #2: No

5. Review Comments to the Author

Reviewer #1: In the manuscript PONE-D-21-31549, entitled “Association of Viral Kinetics, Infection History and Plasma Leakage among Indonesian Dengue Infected Patients” authors describe a small prospective observational study including 48 cases of dengue. The manuscript brings relevant information, although not new, that viral load is correlated with severity (plasma leakage/DHF) specially in primary dengue. The manuscript could be suitable for publication once small modifications are done.

-Page 4, line 107: Authors should better describe the methodology and modifications of Lanciotti’s two-step reverse transcriptase PCR used to determine serotype. Which primers were used?

-Page 6: What is the asterisk in Table 1?

-Replace the term copy/mL for copies/mL

-Table 3 legend should briefly describe method used for serotyping and statistical analysis.

-Authors should briefly describe the dengue grades classification based on the 1997 WHO classification and include number of participants of each subgroup. Tables should differentiate hemorrhagic manifestation withing the groups as well as DHF I and II. All DHF patients displayed plasma leakage signs were classified as DHF since day 1? Table 3 should also include DF, DHF-DSS classification.

-Authors should standardize group names: at some parts of the manuscript groups are called “with plasma leakage” and “without plasma leakage” and in other parts of the text/figures groups are called “plasma leakage” and “no plasma leakage” or non-plasma leakage.

Some sentences are misleading, see examples below:

-Page 7, Lines 170-171 “Between patients with and without plasma leakage, the viral load was significantly different on day 4 (P=0.010) of fever.” Should be rewritten to be more specific:

Suggestion: “When comparing patients under primary infection with versus without plasma leakage, the viral load was significantly higher on day 4 (P=0.010) of fever in the leakage group.”

-Page 7, Lines 170-173: “In patients with secondary infection, the viral load levels of plasma leakage patients were also higher compared to patients without plasma leakage especially on day 1, 3, and 4. However, we did not find a significant difference of viral load in patients with secondary infection (Figure 2B).”

Suggestion: “In patients with secondary infection, the viral load levels of plasma leakage patients were also higher compared to patients without plasma leakage especially on days 1, 3, and 4 although no significant differences were found (Figure 2B).”

-Page 9, line 211. Authors mentioned one case of coinfection in the study, however it is not clear whether the co-infection case was included in the data since table 2 show 28 cases of no leak and 20 cases of leak.

-Language must be revised throughout the text. Some sentences are broken and need to be rewritten. See example below:

Page 10, Lines236-237: “While this may imply more rapid viral clearance in secondary infection, further investigation is still needed as a different marker compared to our study is used.”

Reviewer #2: This study from Nainggolan et al evaluate the association of viral kinetics, infection History and plasma leakage among dengue patients in Indonesia.

The manuscript in poorly written with poor description of the methods which makes it difficult to understand.

The more specific comments in whole is mentioned here.

The authors need to clarify why 1997 WHO criteria used for the classification of dengue infection?

I gather that this small sample (48) of patients were collected in 2010 and submitted for evaluation more than 10 years later. I am unable to understand why this significant delay in submission and weather some of the tests were done in freezed samples.

It also appears that almost all admissions were within two days of the onset of fever and the number of patients developed DHF seems to be unusually high (41.7%) ? I am puzzled with this finding especially in this younger population (mean age 24.9). Other laboratory parameters like AST and AST levels should have mentioned if available and unavailability of those primary data questions the suitability for further evaluation.

The dengue antibody testing (IgM and IgG) needs more details. What test kits were used and how the evaluation was done.

More in-depth analysis would be preferred in the results.

There were many topographical errors which are too many to be listed.

6. PLOS authors have the option to publish the peer review history of their article (what does this mean?). If published, this will include your full peer review and any attached files.

Reviewer #1: No

Reviewer #2: No

---

## [Author Response · Author response to Decision Letter 0]

23 Nov 2022

Reviewer #1

In the manuscript PONE-D-21-31549, entitled “Association of Viral Kinetics, Infection History and Plasma Leakage among Indonesian Dengue Infected Patients” authors describe a small prospective observational study including 48 cases of dengue. The manuscript brings relevant information, although not new, that viral load is correlated with severity (plasma leakage/DHF) specially in primary dengue. The manuscript could be suitable for publication once small modifications are done. 

-Page 4, line 107: Authors should better describe the methodology and modifications of Lanciotti’s two-step reverse transcriptase PCR used to determine serotype. Which primers were used?

Answer: Thank you for the question. A two-step RT-PCR with slight modification was conducted. We have put more detailed information concerning this in the materials and methods section.

-Page 6: What is the asterisk in Table 1?

Answer: Thank you for the question. We have inserted more explanation regarding symbols used below the tables.

-Replace the term copy/mL for copies/mL

Answer: Thank you for your input. We have changed the related terms.

-Table 3 legend should briefly describe method used for serotyping and statistical analysis. 

Answer: Thank you for your input. We have inserted the method used for serotyping and analysis below the table.

-Authors should briefly describe the dengue grades classification based on the 1997 WHO classification and include number of participants of each subgroup. Tables should differentiate hemorrhagic manifestation withing the groups as well as DHF I and II. All DHF patients displayed plasma leakage signs were classified as DHF since day 1? Table 3 should also include DF, DHF-DSS classification.

Answer: Thank you for the suggestion. All patients displayed plasma leakage signs throughout the course of study were classified as DHF. 

-Authors should standardize group names: at some parts of the manuscript groups are called “with plasma leakage” and “without plasma leakage” and in other parts of the text/figures groups are called “plasma leakage” and “no plasma leakage” or non-plasma leakage.

Answer: Thank you for your input. We have made some corrections regarding this.

Some sentences are misleading, see examples below:

-Page 7, Lines 170-171 “Between patients with and without plasma leakage, the viral load was significantly different on day 4 (P=0.010) of fever.” Should be rewritten to be more specific:

Suggestion: “When comparing patients under primary infection with versus without plasma leakage, the viral load was significantly higher on day 4 (P=0.010) of fever in the leakage group.”

Answer: Thank you for your suggestion. We have made some corrections regarding this.

-Page 7, Lines 170-173: “In patients with secondary infection, the viral load levels of plasma leakage patients were also higher compared to patients without plasma leakage especially on day 1, 3, and 4. However, we did not find a significant difference of viral load in patients with secondary infection (Figure 2B).” 

Suggestion: “In patients with secondary infection, the viral load levels of plasma leakage patients were also higher compared to patients without plasma leakage especially on days 1, 3, and 4 although no significant differences were found (Figure 2B).” 

Answer: Thank you for your suggestion. We have made some corrections regarding this.

-Page 9, line 211. Authors mentioned one case of coinfection in the study, however it is not clear whether the co-infection case was included in the data since table 2 show 28 cases of no leak and 20 cases of leak.

Answer: Thank you for your question. The one case of coinfection was included in the data.

-Language must be revised throughout the text. Some sentences are broken and need to be rewritten. See example below: 

Page 10, Lines236-237: “While this may imply more rapid viral clearance in secondary infection, further investigation is still needed as a different marker compared to our study is used.”

Answer: Thank you for your input. We have revised the language used throughout the text.

Reviewer #2

This study from Nainggolan et al evaluate the association of viral kinetics, infection History and plasma leakage among dengue patients in Indonesia.

The manuscript in poorly written with poor description of the methods which makes it difficult to understand.

The more specific comments in whole is mentioned here.

The authors need to clarify why 1997 WHO criteria used for the classification of dengue infection?

Answer: Thank you for your question. We understand that WHO had published newer criteria concerning dengue classification, however we used 1997 WHO criteria due to the detailed definition of plasma leakage provided in this specific criterion.

I gather that this small sample (48) of patients were collected in 2010 and submitted for evaluation more than 10 years later. I am unable to understand why this significant delay in submission and weather some of the tests were done in freezed samples.

Answer: Thank you for your concern. Tests were not done in freezed samples. Delay in submission was due to personal reasons andin the process of writing the manuscript.

It also appears that almost all admissions were within two days of the onset of fever and the number of patients developed DHF seems to be unusually high (41.7%) ? I am puzzled with this finding especially in this younger population (mean age 24.9). Other laboratory parameters like AST and AST levels should have mentioned if available and unavailability of those primary data questions the suitability for further evaluation.

Answer: Thank you for your concern. We have inserted the AST and ALT levels in the Table 1.

The dengue antibody testing (IgM and IgG) needs more details. What test kits were used and how the evaluation was done.

Answer: Thank you for the input. More detailed methods have been inserted in the material and methods section.

More in-depth analysis would be preferred in the results.

There were many topographical errors which are too many to be listed.

---

## [Editor Report · Decision Letter 1]

27 Feb 2023

PONE-D-21-31549R1Association of Viral Kinetics, Infection History, NS1 Protein with Plasma Leakage among Indonesian Dengue Infected PatientsPLOS ONE

Dear Dr. Nainggolan,

Thank you for submitting your manuscript to PLOS ONE. After careful consideration, we feel that it has merit but does not fully meet PLOS ONE’s publication criteria as it currently stands. Therefore, we invite you to submit a revised version of the manuscript that addresses the points raised during the review process. The technical detail that you have added in your revision is satisfactory, however the writing needs further revision to resolve lack of clarity. Please be sure to have your manuscript edited by a native speaker of the English language who is a scientist. If necessary, you might employ a scientific editing service.

We look forward to receiving your revised manuscript.

Kind regards,

Ulrike Gertrud Munderloh, Ph.D.

Academic Editor

PLOS ONE
---

## [Author Response · Author response to Decision Letter 1]

13 Apr 2023

The technical detail that you have added in your revision is satisfactory, however the writing needs further revision to resolve lack of clarity. Please be sure to have your manuscript edited by a native speaker of the English language who is a scientist. If necessary, you might employ a scientific editing service.

Answer: Respected reviewers and editors, thank you for your inputs. We have reviewed our manuscript to improve the writing and have requested a native speaker to review our manuscript. May you find it satisfactory for publication in your journal.

Answer: Thank you for the input. We have reviewed our reference list and made some corrections. We have not found any retracted references however some adjustments to the reference list were made as follows:

• All DOI identifiers, database’s unique identifiers have been removed to comply to samples of standard journal article provided by International Committee of Medical Journal Editors (ICMJE)

• Reference number 2: we have changed Asia WROFS-E to World Health Organization

---

## [Editor Report · Decision Letter 2]

17 Apr 2023

Association of Viral Kinetics, Infection History, NS1 Protein with Plasma Leakage among Indonesian Dengue Infected Patients

PONE-D-21-31549R2

Dear Dr. Nainggolan,

We’re pleased to inform you that your manuscript has been judged scientifically suitable for publication and will be formally accepted for publication once it meets all outstanding technical requirements.

Kind regards,

Ulrike Gertrud Munderloh, Ph.D.

Academic Editor

PLOS ONE
---

## [Editor Report · Acceptance letter]

24 Apr 2023

PONE-D-21-31549R2 

Association of Viral Kinetics, Infection History, NS1 Protein with Plasma Leakage among Indonesian Dengue Infected Patients 

Dear Dr. Nainggolan:

I'm pleased to inform you that your manuscript has been deemed suitable for publication in PLOS ONE. Congratulations! Your manuscript is now with our production department. 

Kind regards, 

on behalf of

Dr. Ulrike Gertrud Munderloh 

Academic Editor

PLOS ONE